# Heat Enhancement Effectiveness Using Multiple Twisted Tape in Rectangular Channels



**M. Ziad Saghir *** , **Ayman Bayomy and Md Abdur Rahman**

Department of Mechanical and Industrial Engineering, Ryerson University, Toronto, ON M5B 2K3, Canada; ayman.bayomy@ryerson.ca (A.B.); md.rahman@ryerson.ca (M.A.R.)
* Correspondence: zsaghir@ryerson.ca

**Abstract:** Heat enhancement and heat removal have been the subject of considerable research in the energy system field. Flow-through channels and pipes have received much attention from engineers involved in heat exchanger design and construction. The use of insert tape is one of many ways to mix fluids, even in a laminar flow regime. The present study focused on the use of different twisted tapes with different pitch-to-pitch distances and lengths to determine the optimum design for the best possible performance energy coefficient. The results revealed that twisted tape of one revolution represented the optimal design configuration and provided the largest Nusselt number. The length of the tape played a major role in the pressure drop. The results revealed that the insertion of a shorter twisted tape can create mixing while minimizing the changes in the pressure drop. In particular, the best performance evaluation criterion is found for a short tape located towards the exit of the channel. The highest performance energy coefficient was obtained for the half-twisted tape for a Reynolds number varying between 200 and 600.

**Keywords:** rectangular channels; twisted tape inserts; laminar flow; forced convection; water; heat enhancement; friction factor; performance evaluation criteria





## 1. Introduction

The development of a heat exchanger is one of the top priorities in engineering applications. A high heat performance heat exchanger is critical for many current engineering applications. Nowadays, the focus is on the design of compact high-performance heat exchangers. In most applications, water is used as the circulating fluid, but it has several limitations; therefore, nanofluids have been introduced as a replacement, and research has shown that they generate a 6% heat enhancement over water [1–5]. Nanofluids consist of a fluid (usually water) mixed with metallic nanoparticles up to 100 nm in diameter in concentrations between 0.1 and 10%vol. Unfortunately, this improved performance comes with a negative setback: a higher pressure drop in the system. In addition, nanoparticles may become corrosive and destroy the heat exchanger, especially in a smaller-scale design. The viscous and thermal boundary layers reduce the heat transfer from the system to the cooling fluid. One approach to overcoming this problem is to insert different types of tape inside the tube or channel to destroy the boundary layer, thereby allowing mixing while remaining in the laminar regime. The tape can also enhance heat performance without changing the circulating fluid, which is mostly water. Despite this, pressure drops still occur. The objective of this paper was to identify a new insert tape capable of heat enhancement while only increasing the pressure drop slightly to keep it within an acceptable range.

Feng et al. [6] experimentally investigated the performance of a micro-mini-channel in the presence of twisted tape. Different tape lengths and twist ratios were numerically investigated. When water was used as the flowing fluid, the experimental results revealed that pressure drops were higher compared to those in a plain channel. The thermal heat enhancement increased along with increases in the twist ratio and tape length. The optimal insert had a length of one (i.e., along the entire channel) with a twist ratio of three. An

empirical relationship between the pressure drop and the heat convection coefficient was proposed as a function of the twist ratio, Reynolds number, Prandtl number and insert length. The authors stated that the average heat transfer coefficient, pressure drops per unit length and thermal enhancement factor increased with decreases in the twist ratio or increased with the length of the twisted tape.

Ahmad et al. [7] numerically and experimentally investigated the effectiveness of using twisted tape in a tube. The flow was turbulent and the flowing fluid was a nanofluid containing silicon carbide and aluminum oxide nanoparticles. Three different twisted tapes were used. The first had a constant pitch ratio, the second had an increasing pitch ratio and the third had a decreasing pitch ratio. The results revealed enhanced heat removal in the presence of the three different twisted tapes. The results also revealed that thermal performance was affected at higher Reynolds numbers due to the change in pressure drop. The use of twisted tape generated a stronger swirling flow at the inlet of the tube and a lower pressure drop near the exit of the pipe.

Twisted tapes of different shapes and types have been used in circular pipes by researchers. Eiamsa-ard et al. [8] used an overlapped twisted tape for the different ratios, and heat enhancement with a smaller overlapped twist ratio was noticeable. Bahiraei et al. [9] used a single-twisted tape, twin co-twisted tapes and twin counter-twisted tapes, which caused swirling and co-swirling flows. Zheng et al. [10] used dimpled twisted tapes with a nanofluid. The results revealed heat enhancement as well as moderate flow resistance. Li et al. [11] used helical-shaped twisted tapes and observed that thermal entropy generation decreased with the rise in the pitch ratio, while frictional entropy generation increased with a high height ratio of the twisted tapes.

Performance evaluation criteria (PEC) for a circular pipe were developed and explained by Webb [12]. The PEC were presented for the four different designs and a relationship between the Nusselt number and the friction coefficient was proposed. Ray and Date [13] extended the work of Webb by studying the friction characteristics and heat transfer characteristics of a twisted tape in a rectangular channel. They observed a correlation between the friction coefficient and the Reynolds number, and a pitch ratio was proposed for the laminar and turbulent flows. Similarly, a relationship between the Nusselt number, Reynolds number and pitch ratio was also proposed for the laminar and turbulent flows.

Other researchers investigated different insert types in laminar and turbulent regimes. Man et al. [14] experimentally investigated the heat transfer and friction factor of a dual-pipe heat exchanger for a single-phase forced convection with clockwise and counter clockwise alternations of the twisted tape. He et al. [15] experimentally investigated heat transfer enhancement in a tube in the presence of cross hollow twisted tape inserts. This unique design was in good agreement with the numerical results with a performance evaluation criterion below one. Liu et al. [16] used a coaxial cross-twisted tape in a laminar flow. A large performance evaluation criterion was found. Another unique design was proposed in this reference. Hong et al. [17] proposed overlapping multiple twist tapes in a tube with a turbulent flow. Chai et al. [18] studied the thermal and hydraulic performance for a laminar flow in a microchannel with fan-shaped ribs on the sidewalls. A performance criterion of 1.33 was obtained numerically.

Based on a review of the literature, the goal of the present paper was to numerically investigate the use of a twisted tape in the channel. Different twist ratios and lengths were used. One of the unique aspects of this research is that the twisted tape did not fill the entire channel in some cases, and the twisted tape did not touch the channel walls, which left a small gap for the fluid to pass without going through the twisted tape. The twisted tape located in the channel created a swirl while remaining in the laminar regime. This mixing reduced the viscous and thermal boundary layers, thereby improving the heat transfer. The novel approach is the twist tape location. Two short twist tapes were located near the end of the channel. Section 2 presents the problem and the numerical model. Section 3 presents the results and discussion, and Section 4 presents the conclusions.

## 2. Problem Description

In the present study, efforts were made to investigate the use of different inserts in a rectangular channel. The primary goal was to enhance the heat removal without increasing the pressure drop. Seven different insert shapes were investigated. Figure 1 presents the model under investigation. This model has been numerically and experimentally investigated by different researchers [1–5]. In the present study, the channels used were plain channels with a twisted insert in the channels. Previous studies have numerically and experimentally demonstrated that nanofluids lead to better head enhancement than water [6]. The downfall is that nanofluids create a larger pressure drop compared to water. This study aimed to find a way to improve heat extraction while using water as a cooling fluid.

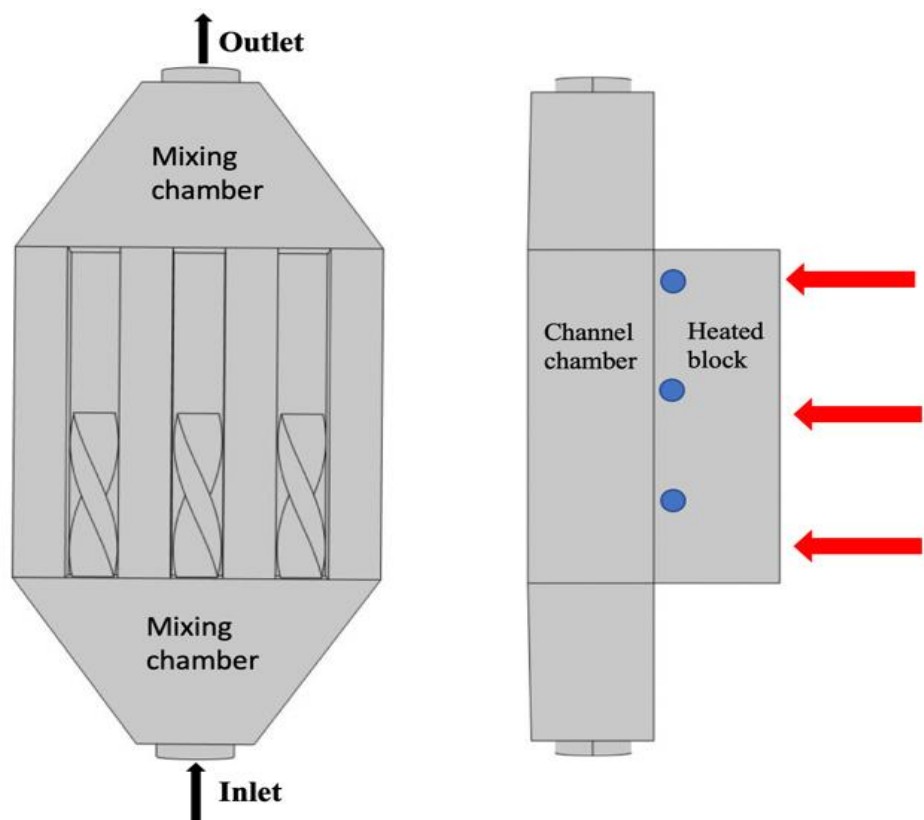

**Figure 1.** Problem description.

Two mixing chambers were present at the inlet and outlet and the flow entered at a certain temperature, $T_{in}$, and flow rate. The temperature was measured 1 mm below the interface (see the blue dots) and the thermocouple was located in the middle of the heated block, which was made of aluminum. The channel had a width of 0.00535 m and a height of 0.0127 m. The channel chamber was a square of 0.0375 m in size, made of aluminum as well. The red arrows in Figure 1 show the heated element location where the heat flux was applied. The flow entered from the inlet with at a certain velocity, $u_{in}$, and temperature, $T_{in}$. A free boundary was applied at the outlet, as shown in Figure 1. All surrounding surfaces were insulated; thus, no heat was lost. A heat flux was applied at the bottom plate where the red arrow is shown in Figure 1. A total of seven different inserts were used, mostly constructed from aluminum, with either one, three or five revolutions. Two additional inserts with either a half revolution or a three-quarter revolution were used in our analysis. These two short inserts were located in different places. The first inserts were located at the entrance of the flow and identified as 3 half revolutions and left three-quarter revolution inserts. The final two configurations were similar in size to the previous one, but

located at the end of the channel. These were identified as right half revolution and right three-quarter revolution inserts. Figure 2 displays the one-revolution, three-revolution and five-revolution twisted tape located inside the channels, and Figure 3 displays the left half revolution and the right half revolution inserts.

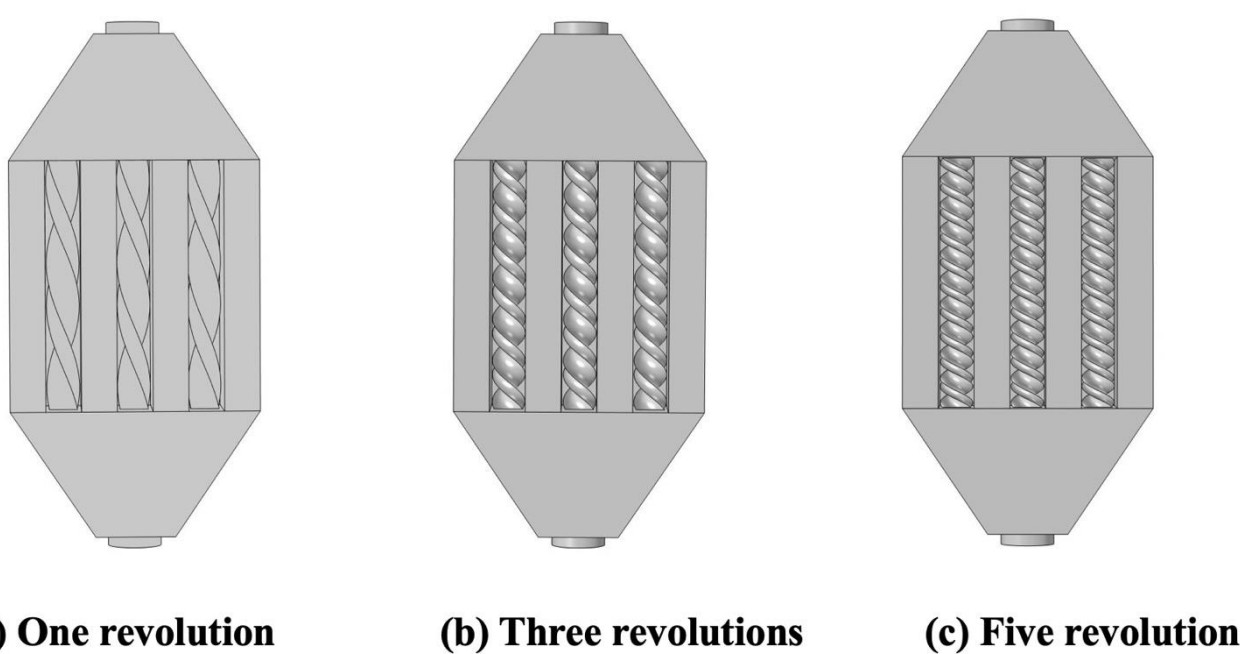

**(a) One revolution**  **(b) Three revolutions**  **(c) Five revolutions**

**Figure 2.** Different insert models.

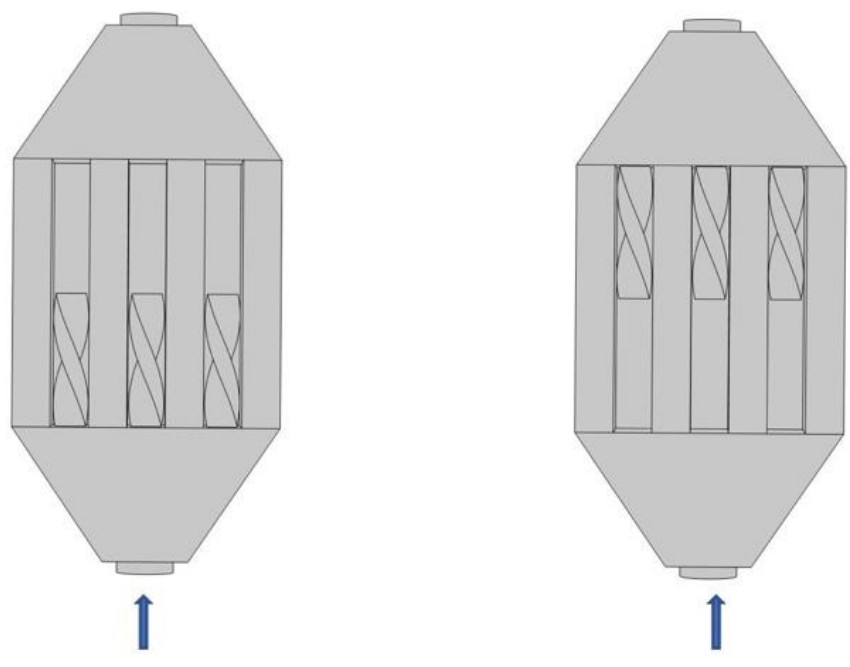

**Left half revolution**  **Right half revolution**

**Figure 3.** Short twisted tape inserts locations.

The purpose of these inserts was to avoid a large pressure drop. For that reason, a gap with no obstruction was included at the top and bottom of the channel, allowing the flow to move freely at those locations. Studies have shown that the viscous boundary layer and

the thermal boundary layer reduce the heat enhancement in a plain channel. It was hoped that by installing the inserts, the flow going through the insert will create mixing and a swirl flow, leading to a reduction in these two boundary layers.

Another approach used in this study involved shortening the insert to allow some mixing at the beginning of the channel, aiming at reducing the boundary layers in the plain section of this channel. Two different forced convection cases were investigated, as shown in Figure 3. In the first case, the length of the insert was half of the channel length, and in the second case, the insert length was three-quarters of the channel length. In addition, two approaches were investigated for each case. In the first approach, the twisted tape location started at the inlet, and in the second approach, the short twisted tape ended at the exit section, thus allowing the flow in the first approach to interact first with the insert; in the second approach, the flow will start interacting with the plain channel and then pass through the insert. The main reason for the location of the short insert is to be able to destroy the boundary layer build-up along the channel towards the end, hence improving the heat extraction. Figure 3 presents the two approaches for a half-twisted tape. A cross-section demonstrating the location of the insert and the gap left at the bottom and at the top of the channel is shown in Figure 4.

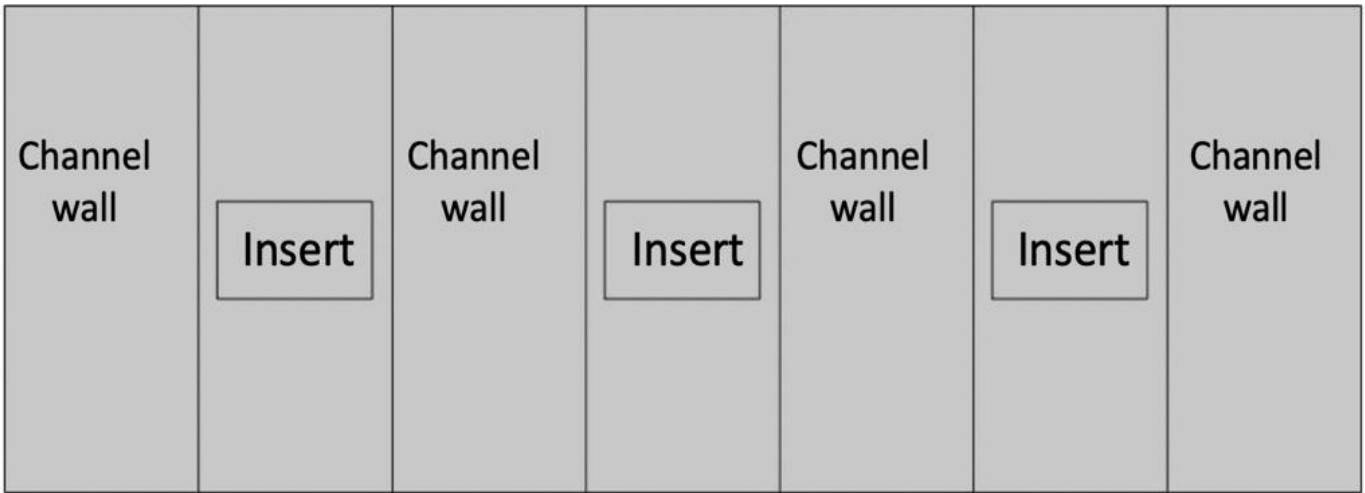

**Figure 4.** Cross-sectional area of the three channels.

The wall thickness of the channel was identical to the channel thickness (0.00535 m) and the height was 0.0127 m. The insert width was 0.0025 m, creating identical gaps from the bottom of the channel to the insert boundary (0.0014 m) and at the top of the channel. As previously mentioned, this new model aimed to allow the fluid to move freely at the bottom and top gaps while forcing some mixing through the twisted tape. In the past, researchers [6] filled the channel with a twisted tape and observed a pronounced pressure drop. Here, it is interesting to study the presence of a combination of a plain channel with the twisted tape section and to study their interaction.

## 3. Finite Element Formulations and Boundary Conditions

The full Navier–Stokes equation together with the energy equation were solved numerically using the finite element code COMSOL. In addition, the heat conduction equation was also solved for the solid part of the model, which is mainly the channel walls as well as the heated block. The flow is assumed to be Newtonian in the laminar regime and steady state. All physical properties of the fluid are assumed to be constant. Below, we present the formulation in a non-dimensional form.



### 3.1. Fluid Flow Formulation

The following non-dimensional parameter was used:

$$X = \frac{x}{D}, \ Y = \frac{y}{D}, \ Z = \frac{z}{D}, \ U = \frac{u}{u_{in}}, \ V = \frac{v}{u_{in}}, \ W = \frac{w}{u_{in}}, P = \frac{pD}{\mu u_{in}}, \ \theta = \frac{(T - T_{in})k}{q''D} \quad (1)$$

The following non-dimensional terms were obtained:

$$Re = \frac{\rho u_{in} D}{\mu}, \ Pr = \frac{cp.\mu}{k} \quad (2)$$

where $u_{in}$ is the velocity at the inlet, as shown in Figure 1; the characteristic length D is equal to 0.01897 m; and Re and Pr are the Reynolds number and the Prandtl number of the fluid, respectively, which is water in the present study. The physical properties of the water were taken from the literature, assumed at room temperature.

The full Navier–Stokes equation in three dimensions is as follows:

X direction,

$$Re\left[U\frac{\partial U}{\partial X} + V\frac{\partial U}{\partial Y} + W\frac{\partial U}{\partial Z}\right] = -\frac{\partial P}{\partial X} + \left[\frac{\partial^2 U}{\partial X^2} + \frac{\partial^2 U}{\partial Y^2} + \frac{\partial^2 U}{\partial Z^2}\right] \quad (3)$$

Y direction,

$$Re\left[U\frac{\partial V}{\partial X} + V\frac{\partial V}{\partial Y} + W\frac{\partial V}{\partial Z}\right] = -\frac{\partial P}{\partial Y} + \left[\frac{\partial^2 V}{\partial X^2} + \frac{\partial^2 V}{\partial Y^2} + \frac{\partial^2 V}{\partial Z^2}\right] \quad (4)$$

Z direction,

$$Re\left[U\frac{\partial W}{\partial X} + V\frac{\partial W}{\partial Y} + W\frac{\partial W}{\partial Z}\right] = -\frac{\partial P}{\partial Z} + \left[\frac{\partial^2 W}{\partial X^2} + \frac{\partial^2 W}{\partial Y^2} + \frac{\partial^2 W}{\partial Z^2}\right] \quad (5)$$

where U, V and W are the velocities at X, Y and Z in non-dimensional form, respectively.

### 3.2. Energy Formulation

The energy equation for the fluid portion is as follows:

$$RePr\left[U\frac{\partial \theta}{\partial X} + V\frac{\partial \theta}{\partial Y} + W\frac{\partial \theta}{\partial Z}\right] = \left[\frac{\partial^2 \theta}{\partial X^2} + \frac{\partial^2 \theta}{\partial Y^2} + \frac{\partial^2 \theta}{\partial Z^2}\right] \quad (6)$$

The local Nusselt number is known as the ratio of the convective heat coefficient multiplied by the characteristic length over the water conductivity (i.e., $\frac{hD}{k}$); based on the non-dimensional adopted earlier, it becomes the inverse of the temperature. Thus,

$$Nu = \frac{1}{\theta} \quad (7)$$

The Performance Evaluation Criterion, PEC, is taken as the ratio between the studied case and the plain channel. Water is always the current fluid used in our analysis. Thus, the formulation for the PEC number is as follows:

$$PEC = \left[\frac{\overline{Nu_t}}{\overline{Nu_p}}\right] * \left[\frac{\Delta P_p}{\Delta P_t}\right]^{\frac{1}{3}} \quad (8)$$

where $\overline{Nu_t}$ and $\overline{Nu_p}$ are the average Nusselt number for the current case under investigation and the plain channel configuration, respectively. Similarly, the pressure drops $\Delta P_p$ and $\Delta P_t$ are for the case of the plain channel and the case under investigation, respectively [6].

*3.3. Boundary Conditions*

Figure 1 presents the boundary conditions. The fluid enters with a certain temperature, Tin—which, in non-dimensional form, is θ set equal to 1—and an inlet velocity, U, of 1. Three different Reynolds numbers of 200, 400 and 600 were investigated corresponding to an inlet flow rate of 0.033 USGPM ($2.1 \times 10^{-6}$ m$^3$/s), 0.05 USGPM ($4.21 \times 10^{-6}$ m$^3$/s) and 0.1 USGPM ($6.31 \times 10^{-6}$ m$^3$/s), respectively. The flow rate is defined as the product of the inlet velocity multiplied by the sum of the cross-sectional area of the three channels' inlet. The system was heated from the bottom with a heat flux having a value of 1. At the outlet direction, a free boundary was applied. The entire external surfaces are assumed to be insulated to eliminate heat loss to the atmosphere.

## 4. Mesh Sensitivity Analysis and Convergence Criteria

The mesh sensitivity was examined in order to determine the optimal mesh required for the analysis. In the table below, we demonstrate the different mesh sensitivities that were investigated following the terminology used by COMSOL software.

The mesh levels that COMSOL supports and the element numbers for each mesh level are shown in Table 1. The average Nusselt number was evaluated at 1 mm below the interface in the aluminum block, and the results are shown in Figure 5a.

**Table 1.** Mesh information for different levels of meshing.

| | |
|---|---|
| Extremely coarse | 28,467 domain elements, 5622 boundary elements, 834 edge elements |
| Extra coarse | 46,572 domain elements, 7987 boundary elements, 1040 edge elements |
| Coarser | 90,505 domain elements, 12,296 boundary elements, 1335 edge elements |
| Coarse | 220,343 domain elements, 23,474 boundary elements, 1879 edge elements |
| Normal | 508,380 domain elements, 42,191 boundary elements, 2612 edge elements |
| Fine | 1,252,080 domain elements, 77,289 boundary elements, 3605 edge elements |

It is evident that a normal mesh level will be suitable to be used in the COMSOL model. Figure 5b presents the finite element mesh used in our simulation.

Different approaches exist in COMSOL to tackle the convergence criteria. In this particular model, the default solver used was the segregated method. Details about this approach can be found in any finite element textbook. The convergence criterion is clearly explained in the COMSOL manual. In brief, the convergence criterion was set as follows: at every iteration, the average relative errors of U, V, W, P and θ were computed. These were obtained using the following relation:

$$R_c = \frac{1}{n \cdot m} \sum_{i=1}^{i=m} \sum_{j=1}^{j=n} \left| \frac{\left(F_{i,j}^{s+1} - F_{i,j}^{s}\right)}{F_{i,j}^{s+1}} \right| \tag{9}$$

where F represents one of the unknowns, viz. U, V, W, P or θ; s is the iteration number and (i, j) represents the coordinates on the grid. Convergence is reached if $R_c$ for all the unknowns is below $1 \times 10^{-6}$ in two successive iterations. For further information on the detailed solution method, the reader is referred to the COMSOL software manual [19]. In addition, the model was validated against experimental data for an insert made of a porous medium. Different publications [1,4] have proven the accuracy of the numerical model.

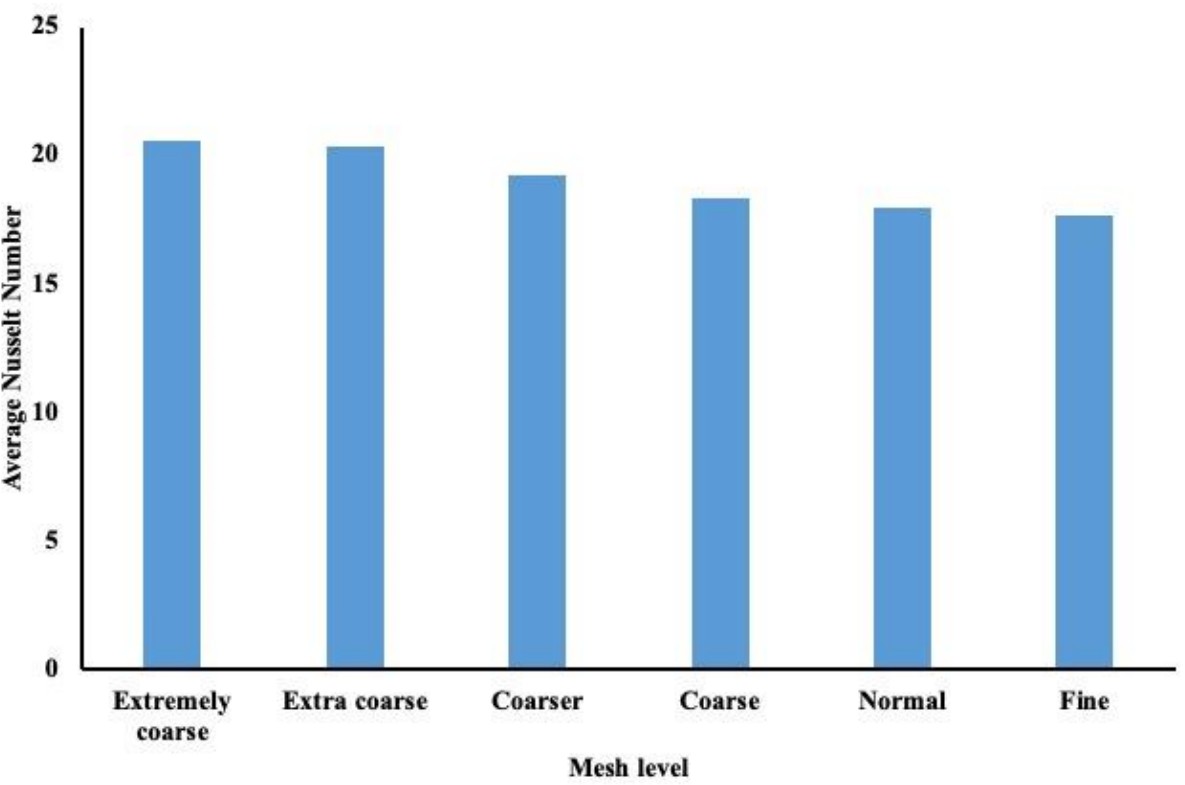

(**a**) Mesh sensitivity analysis

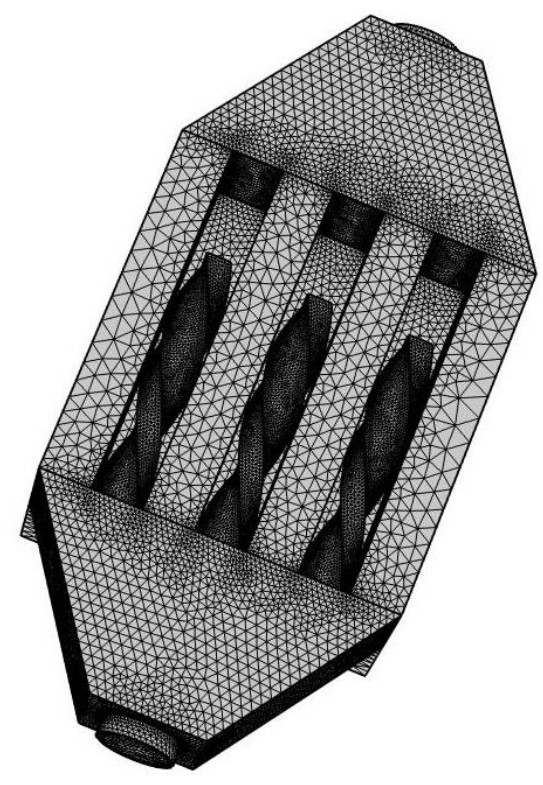

(**b**) Finite element model

**Figure 5.** Finite element analysis.

## 5. Results and Discussion

There are different advantages to the use of twisted tape in a plain channel. The literature review revealed that twisted tape enhances heat removal at the expense of the pressure drop. Thus, one may obtain a higher Nusselt number in the presence of twisted tape but end up with lower performance evaluation criteria (PEC) due to the large pressure drop. In the proposed design, a fluid gap was maintained between the channel of four walls and the twisted tape having different lengths and positions, which the authors believe could create enough mixing and not affect the pressure drop as in the plain channel configuration. Modest mixing may reduce the thermal and viscous boundary layers, thus leading to better PEC.

### 5.1. Effect of the Inserts on the Nusselt Number

Two different cases were studied with identical conditions. The first case involved a plain channel and the second case involved twisted tape with one revolution. It is important to note that the revolution occurred along the entire length of the channel. Figure 6 presents the local Nusselt number variation along the flow direction for the three different Reynolds numbers. The Nusselt number was evaluated 1 mm below the middle channel. The results reveal that the increase in the Nusselt number is higher for the twisted tape with one revolution. Based on the figures, one may notice the average increase with one revolution compared to the plain channel of 12% at a Reynolds number equal to 200, of 15% at a Reynolds number equal to 400 and, finally, of 16.76% for a Reynolds number of 600. Due to the nonlinearity of the model, the variation between the Nusselt number increase and the Reynolds number is not linear. It is also important to investigate the PEC for these two systems, which are discussed later in the paper.

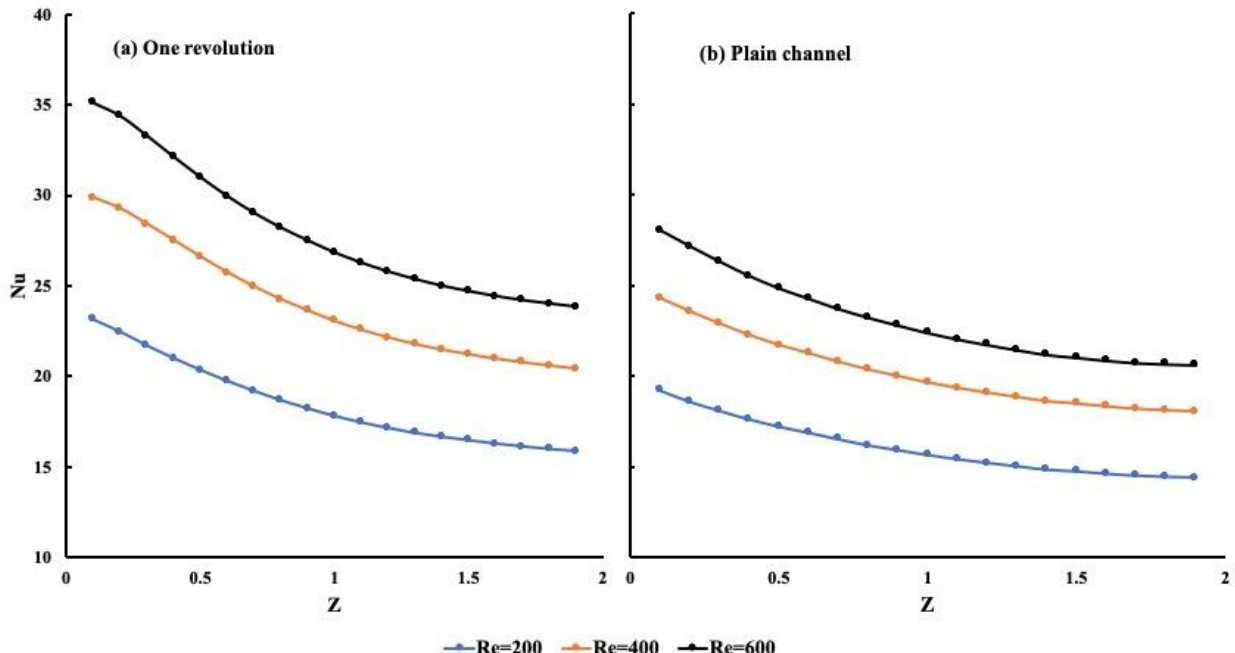

**Figure 6.** Effectiveness of using twisted tape.

The model was further extended and included three cases: (1) the presence of a twisted tape with one revolution over the entire length, (2) the presence of three revolutions and (3) the presence of five revolutions. Figure 7 displays the local Nusselt number for the different Reynolds numbers. The same heating conditions as those seen in Figure 6 were used. The results revealed a minimal increase in the average local Nusselt number as the number of revolutions increased. Compared to the case with one revolution (Figure 7a), the cases with three revolutions (Figure 7b) and five revolutions (Figure 7c) showed a difference of

0.36% and 0.1% for a Reynolds number of 200, −0.18% and −0.5% for a Reynolds number of 400 and 0.5% and −0.3% for a Reynolds number of 600, respectively. The negative sign indicates a decrease in the average Nusselt number as the number of revolutions in the twisted tapes increased. The variation in the local Nusselt number was very close for all three cases. One can conclude from this comparison that the twisted tape with one revolution had the highest average local Nusselt number compared to the other twisted tape configurations.

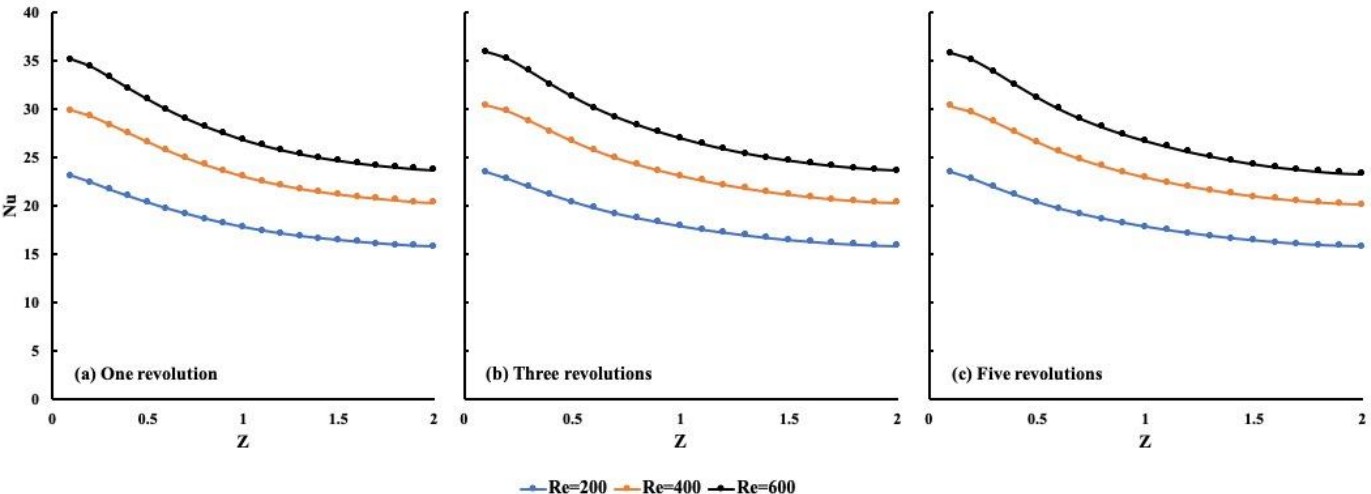

**Figure 7.** Comparison of local Nusselt numbers for different inserts.

The effect of the twist revolution length on the Nusselt number was also investigated. The previous results demonstrated that the twist with one revolution performs better than the plain channel and a similar performance was shown for twisted tape with three revolutions and five revolutions. Figure 8 presents, for the same condition as the other two cases, the importance of one twist revolution length by comparing it with half and three-quarter revolutions. The location of this short twisted tape started at the flow inlet and it was thus identified as left twisted tape. We also considered how one can increase the Nusselt number by reducing the pressure drop to the minimum. In Figure 8, a comparison between one revolution and left half and left three-quarter revolutions is presented. It shows a mixing from the inlet and then plain channel flow as it leaves the channel. As shown in Figure 8, a minimal difference in Nusselt number variation is noticeable.

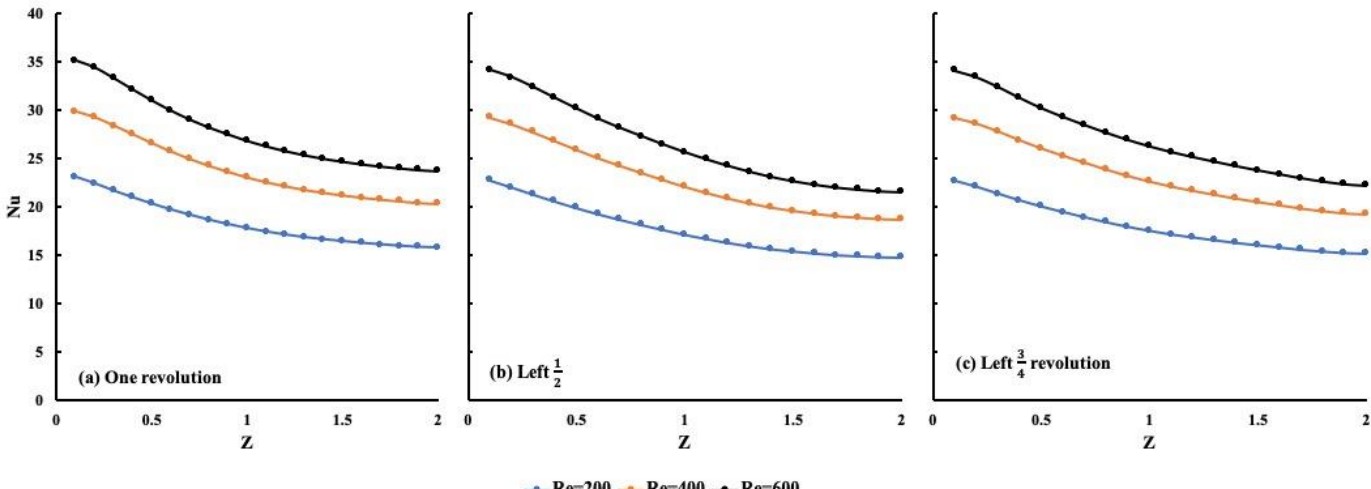

**Figure 8.** Effect of insert length from the flow entry.

For a Reynolds number of 200, the twisted tape with a left half revolution and the twisted tape with a left three-quarter revolution revealed similar Nusselt numbers when compared to the twisted tape with one revolution. The slope of the local Nusselt number variation was greater for the left half twisted tape compared to the other conditions. Thus, if one shortens the twisted tape, no changes in the local Nusselt number variation are noticeable.

The fourth and final configuration under study involved changing the location of the twisted tape by allowing the flow to enter the plain region first and then pass through the twisted tape. Figure 9 displays a comparison between the case with one revolution and the so-called right half revolution and right three-quarter revolution. It is interesting to note that the average Nusselt number was similar for all three configurations, with the highest change of 2.67% for a Reynolds number of 600. However, it is interesting to note that with the change in twisted tape location, the Nusselt number was low at the entrance, i.e., the plain channel, and as the flow hit the twisted tape, the mixing created a higher Nusselt number when compared to the entrance local Nusselt number.

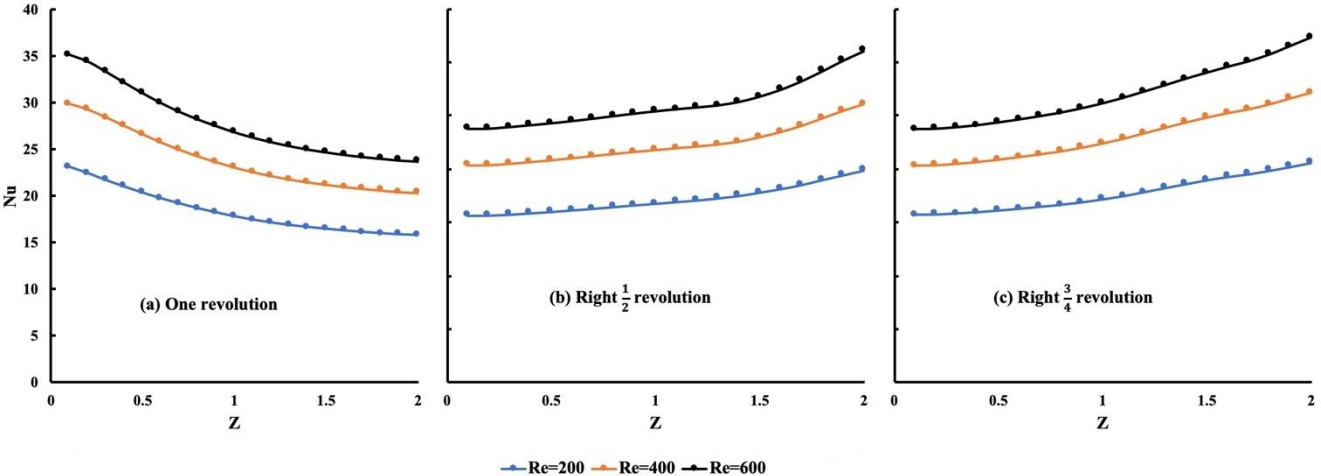

**Figure 9.** Importance of insert location in the channel.

The results indicate that the best configuration is a shortened twisted tape with one revolution.

Figures 10 and 11 present the velocity flow patterns for the one-revolution and half-revolution cases with identical inlet conditions. The results reveal that two different flows were occurring. The first flow was a non-twisted flow at the top and bottom of the tape, as shown with the straight arrows for the velocity. Thus, the fluid escaped by not going through a rotating state. The second flow was a mixing/swirl flow, shown as a red region flowing within the insert twist. As the length of the twisted tape was shortened, a combination of swirl and mixing flows occurred in the beginning of the channel, and a plain channel flow occurred from the middle to the exit of the channel, as shown in Figure 11.

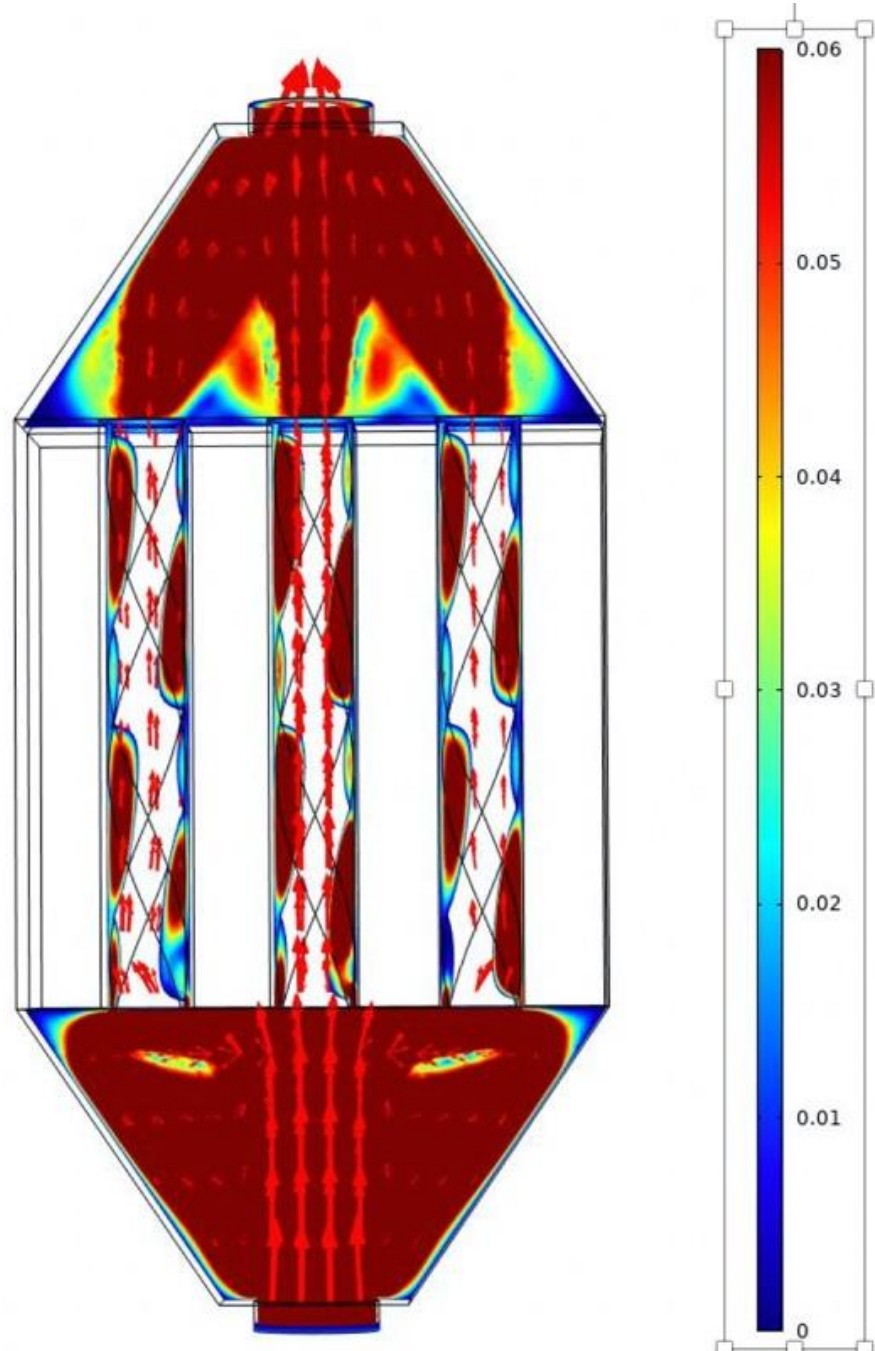

**Figure 10.** Velocity distribution for one-revolution insert case at Re = 200.

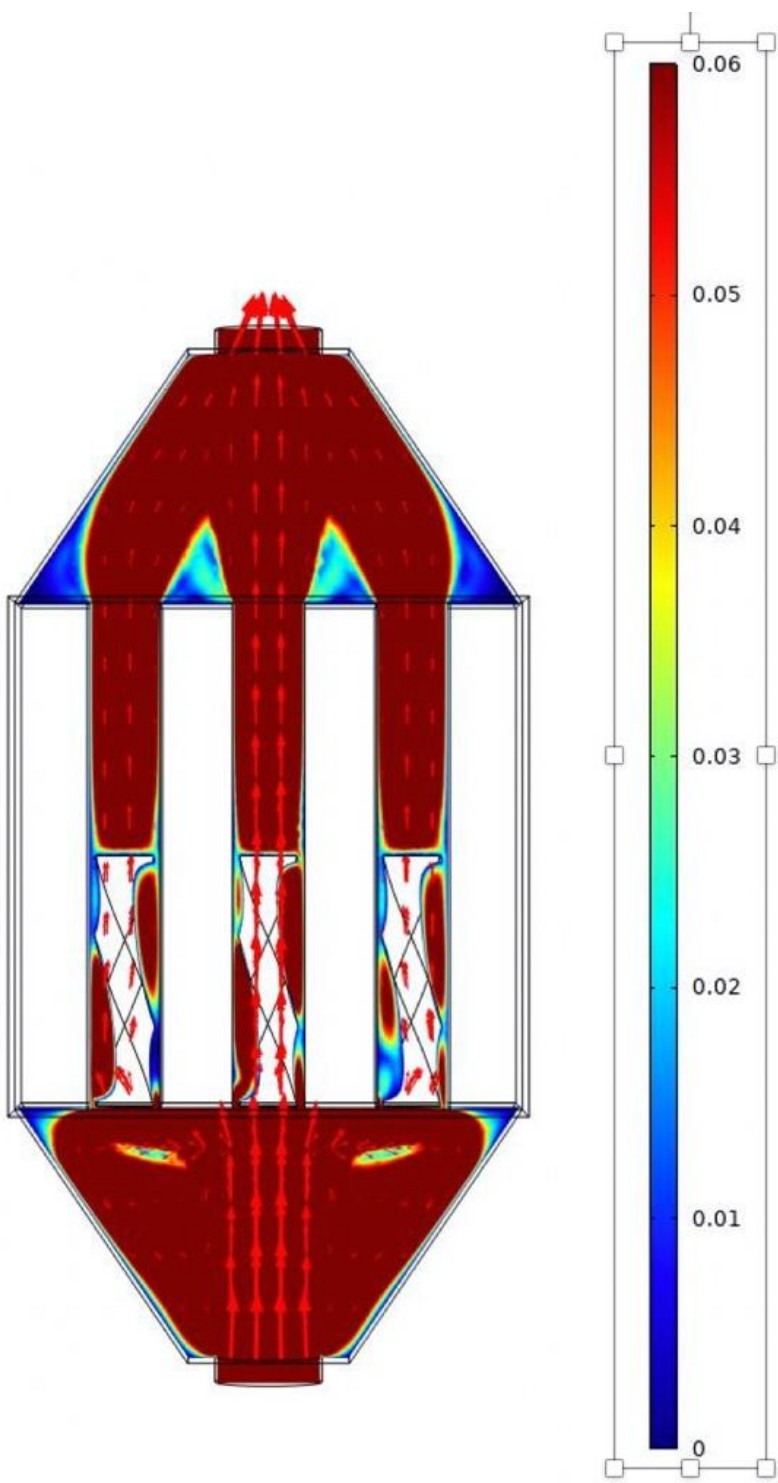

**Figure 11.** Velocity distribution for half-revolution insert case at Re = 200.

### 5.2. Pressure Drop

In the previous section, we discussed the effectiveness of the use of different twisted tape configurations and concluded that the twisted tape with one revolution had a better local Nusselt number than the twisted tape with three or five revolutions. In addition, shortening the one-revolution tape to half and three-quarters had a minor effect on the local Nusselt number. Changing the location of the half and three-quarter tapes also had a minor effect on the local Nusselt number. However, in order to determine the best twisted tape configuration, one must first examine the pressure drop across the channel.

Figure 12 presents the pressure drop variation along the entire system, combining the three channels for all configurations presented earlier. Figure 12a presents the one-revolution case and Figure 12b presents the three-quarter-length twisted tape at two different flow entrances. Finally, Figure 12c is similar to the previous case but for a half-twisted tape. The comparison of all of the configurations revealed that the flow entering from the plain channel side, then crossing the twisted tape, exhibited a lower pressure drop, as shown in Figure 12b,c. The pressure drops varied linearly with the Reynolds number, regardless of the configuration under investigation.

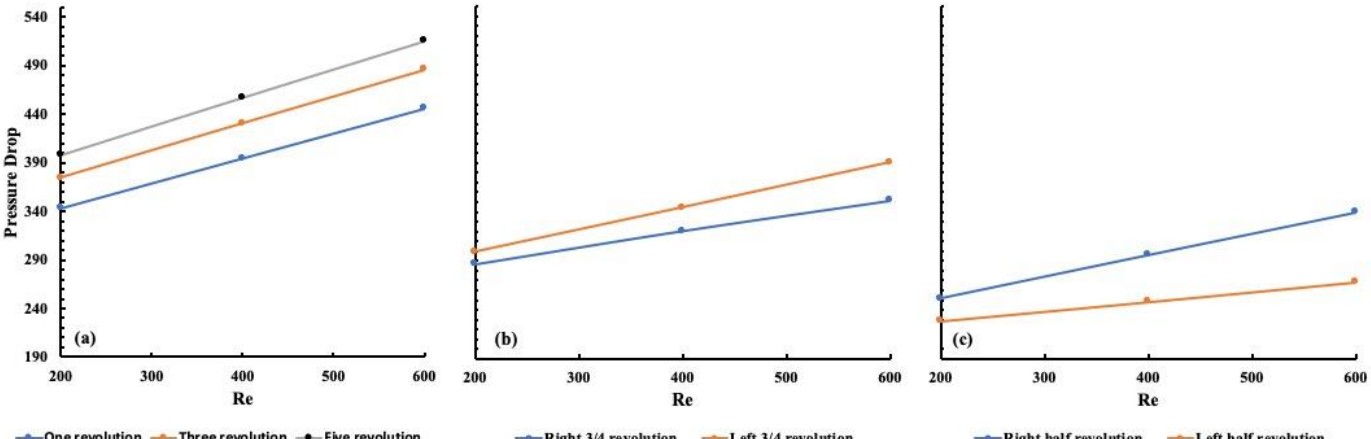

**Figure 12.** Pressure drop along the channel for all inserts.

One may, therefore, conclude that the lowest pressure drop occurs when the flow goes through a plain channel configuration before crossing the twisted tape. Thus, shortening the one-revolution tape is a good approach to minimize the increase in the pressure drop.

Combining the heat transfer effect represented by the local Nusselt number and the flow effect represented by the pressure drop, Figure 13a–c present the PEC number for all of the configurations, including the one-, three- and five-revolution configurations; the three-quarter configuration and the shortened half-revolution configuration, respectively. It is clear in Figure 13c that the right half configuration exhibited the highest PEC number, followed by the right three-quarter twisted tape. Their performance was more pronounced than that of the twisted tape with one revolution. The reason for these findings is the lower pressure drop and similar average Nusselt number to the one-revolution case.

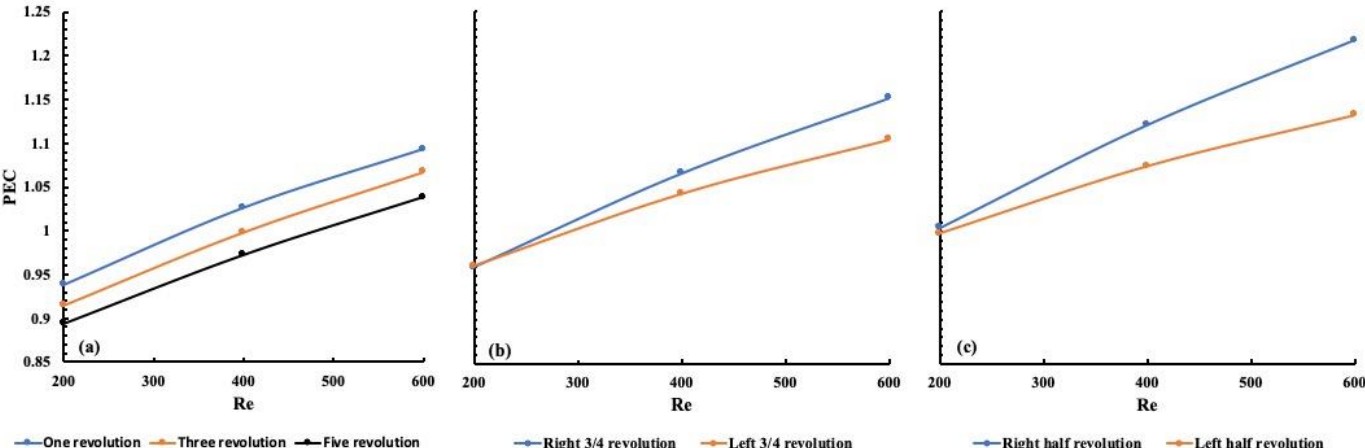

**Figure 13.** PEC number for all inserts at different Reynolds numbers.

## 6. Conclusions

Heat enhancement is a major research objective with many applications in different areas of engineering design. The use of nanofluids has been successful at the expense of pressure drops and the sedimentation of nanoparticles. Twisted tape has been shown to create mixing, even at small Reynolds numbers; however, the pressure drop is still significant. Different twisted tape configurations have, thus, been investigated. The following conclusions were drawn from the findings:

1. Between the three twisted tape configurations (one revolution, three revolutions and five revolutions), the configuration with one revolution revealed the highest performance evaluation criteria when compared to the plain channel configuration.

2. Shortening the twisted tape with one revolution and moving its position provided similar local Nusselt numbers compared to the twisted tape with one revolution.

3. The pressure drop was reduced as the twisted tape with one revolution was shortened.

4. By combining the Nusselt number and the pressure drop, the largest performance evaluation criteria number was obtained for the shorter twisted tape located towards the end of the channel. The location of the half tape plays a major role in heat enhancement. It helps destroy the thermal and fluid boundary layer build-up toward the end of the channel.

**Author Contributions:** Conceptualization, M.Z.S.; methodology, A.B.; software, M.Z.S.; validation, M.A.R., M.Z.S. and A.B.; formal analysis, M.Z.S.; investigation, M.Z.S.; resources, M.Z.S.; writing—original draft preparation, M.Z.S.; writing—review and editing, M.A.R. and A.B.; visualization, M.Z.S.; supervision, M.Z.S.; project administration, M.Z.S.; funding acquisition, M.Z.S. All authors have read and agreed to the published version of the manuscript.

**Funding:** Qatar Foundation, grant number NPRP12S-0123-190011.

**Informed Consent Statement:** Not Applicable.

**Data Availability Statement:** This study did not report any data.

**Acknowledgments:** This research was funded by National Science and Engineering Research Council Canada (NSERC), the Faculty of Engineering and Architectural Science at Ryerson University and the Qatar Foundation, grant number NPRP12S-0123-190011.

**Conflicts of Interest:** The authors declare no conflict of interest.

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
