# Peer review of "Heat Enhancement Effectiveness Using Multiple Twisted Tape in Rectangular Channels"

_fluids, doi:10.3390/fluids6050188_

Round 1
Reviewer 1 Report
The paper submitted considers the impact of twisted tape insert on thermal performance of heat exchanger. The topic is of interest in many applications. Simulations were performed with COMSOL. Different inserts were used in a rectangular channels to investigate the Nusselt number and pressure drop. It was shown in particular how the insert revolution and its length influence on the heat transfer behavior. As pointed out in the introduction, the unique aspects of the research was a gap at the bottom and the top of the channel allowing the fluid to move freely. I generally agree with the comments (the most effective insert is that with one left half revolution) and conclusions. Lastly, I appreciated the calculations of performance energy coefficient (PEC) because this is the most valuable index to compare different solutions in relation to their heat transfer and flow resistance. Nevertheless, there are a number of issues that need to be addressed before paper could be published:
- The authors have performed only numerical investigation and have not verified it with the experiment, even with the literature. Therefore, it is difficult to evaluate the usefulness of the results. I recommend to perform the validation study.
- As pointed out in the manuscript, the unique aspects of the research was a gap at the bottom and the top of the channel, which could create enough mixing and not affecting the pressure drop as in the plain channel configuration. The modest mixing may reduce the thermal and viscous boundary layers, leading to a better PEC. I do not understand, where the gap is located and how its existence influence on the results? Figure 6 shows the impact of the gap on Nusselt number. However, figure presenting the impact of gap on pressure drop or PEC is also required.
- The problem description is not clear and should be improved. According to the text, seven inserts were tested and, as I understand correctly, the inserts were of the following revolution configurations: 1) one, 2) three, 3) five, 4) left half one, 5) right half one, 6) left three-quarter one, 5) right three-quarter one. The revolution configuration should be better described and illustrated. It would be great if one channel filled with insert is presented. I have also the problem with the location of all dimensions. I recommend to add the dimensions in Figure 1.
- I have looked through all References cited in Introduction. References [6, 14] indicate that the smaller the revolution and the insert length, the lower pressure drop and the higher the heat transfer coefficient. Reference [14] considers also one-quarter revolution, for which heat transfer and flow resistance are the most profitable among others. Thus where is the innovation?
- References should be stylistically corrected (description order).
Author Response
Thank you for taking the time to review my paper. Your comments were well appreciated. I have uploaded my rebuttal and a revised manuscript
Best regards
Ziad

Reviewer 2 Report
The authors of the manuscript have investigated the heat enhancement effectiveness using multiple twisted tape in rectangular channels. It is about very interested and important research. I would like to give a couple of suggestions to improve the quality of the manuscript.
One can not see two different forced convection cases in Figure 3. Please, illustrate it better.
Figure 1 partially presents the boundary condition. I suggest you to better present boundary condition in Figure 1. All boundary condition should be clearly presented.
In the Figure 5a there is a big gap between the figure and text.
In it very hard to notice at the figure 10 and 11 these statements: "The results revealed that two different flows were occurring. The first was a smooth flow at the top and bottom of the twisted tape, and the second was a mixing/swirl flow in the middle".
Author Response

(The authors gave the same response as above.)

Reviewer 3 Report
Comments on “Heat Enhancement Effectiveness using Multiple Twisted Tape in Rectangular Channels”
- English requires further proofreading and typos must be eliminated from the paper.
- Introduction needs further enrichment by reviewing more state-of-the-art papers relevant to passive techniques for heat transfer enhancement. Searching the literature, following papers are suggested to be read and used: Efficiency assessment of using graphene nanoplatelets-silver/water nanofluids in microchannel heat sinks with different cross-sections for electronics cooling. Simulation of water/FMWCNT nanofluid forced convection in a microchannel filled with porous material under slip velocity and temperature jump boundary conditions. Two-phase frictional pressure drop with pure refrigerants in vertical mini/micro-channels. Heat transfer of oil/MWCNT nanofluid jet injection inside a rectangular microchannel. Thermal and mechanical design of tangential hybrid microchannel and high-conductivity inserts for cooling of disk-shaped electronic components. Introduce a novel configurationof microchannel andhigh-conductivity insertsfor cooling of disc-shaped electronic components. Configuration and optimization of a minichannel using water–alumina nanofluid by non-dominated sorting genetic algorithm and response surface method.
- A list of assumptions should be added to the paper to improve the reliability of the solution.
- An algorithm of solution in form of a flow chart can help readers understand which equations are solved. It must be added to the paper.
- A robust validation against already verified data or experimental data needs to be conducted and be added to the paper.
- Does this model consider any heat loss to the environment?
- Is there any back pressure and back flow in the heat exchangers?
All in all, the paper can be published once above comments are addressed.
Author Response

(The authors gave the same response as above.)

Round 2
Reviewer 1 Report
The Authors corrected properly their manuscript. All suggestions were taken under consideration.
Author Response
Dear Prof Mamou
Thank you for your clarification. I am attaching a compress file containing the document with highlight and without highlight as well as the revised figure 10 and figure 11 and figure 3. I have changed the caption, added some clarification about the velocity profile and added new figure for figure 10 and figure 11. Figure 3 was modified earlier but I did not upload the revised figure in my rebuttal which is done now.
I hope it meet your expectation
Best regards
Ziad
Reviewer 3 Report
The author did not answer all of my questions accordingly. They just have responded to some parts of the issues and have left some others! I give one more chance to authors to answer ALL of my questions/ comments. If they cannot address all of my questions/ comments, I have no choice just to reject the paper.
Author Response

(The authors gave the same response as above.)
